# From Discovery of Snake Venom Disintegrins to A Safer Therapeutic Antithrombotic Agent

**DOI:** 10.3390/toxins11070372

**Published:** 2019-06-26

**Authors:** Yu-Ju Kuo, Ching-Hu Chung, Tur-Fu Huang

**Affiliations:** 1Department of Medicine, Mackay Medical College, New Taipei City 25245, Taiwan; 2Graduate Institute of Pharmacology, College of Medicine, National Taiwan University, Taipei 10051, Taiwan

**Keywords:** snake venom proteins, disintegrins, antiplatelet agent, arterial thrombosis, angiogenesis, septic inflammation

## Abstract

Snake venoms affect blood coagulation and platelet function in diverse ways. Some venom components inhibit platelet function, while other components induce platelet aggregation. Among the platelet aggregation inhibitors, disintegrins have been recognized as unique and potentially valuable tools for examining cell–matrix and cell–cell interactions and for the development of antithrombotic and antiangiogenic agents according to their anti-adhesive and anti-migration effect on tumor cells and antiangiogenesis activities. Disintegrins represent a family of low molecular weight, cysteine-rich, Arg-Gly-Asp(RGD)/Lys-Gly-Asp(KGD)-containing polypeptides, which inhibit fibrinogen binding to integrin αIIbβ3 (i.e., platelet glycoprotein IIb/IIIa), as well as ligand binding to integrins αvβ3, and α5β1 expressed on cells (i.e., fibroblasts, tumor cells, and endothelial cells). This review focuses on the current efforts attained from studies using disintegrins as a tool in the field of arterial thrombosis, angiogenesis, inflammation, and tumor metastasis, and briefly describes their potential therapeutic applications and side effects in integrin-related diseases. Additionally, novel R(K)GD-containing disintegrin TMV-7 mutants are being designed as safer antithrombotics without causing thrombocytopenia and bleeding.

## 1. Introduction

### 1.1. Role of Platelet Integrin in Thrombosis and Hemostasis 

Integrins, one of the adhesion receptor super-families, play vital roles in regulating cell–matrix and cell–cell interactions. Integrins exist as an α:β heterodimeric complex of transmembrane proteins, and play important roles in platelet aggregation, adhesion, spreading, retraction, migration, angiogenesis, inflammatory reactions, and other biological processes [1,2].

Circulating platelets respond very rapidly to vascular injury. The coverage of the exposed endothelium by activated platelets depends on the recognition of adhesive proteins (i.e., fibrinogen, collagen, von Willebrand factor, and fibronectin) by their specific platelet membrane glycoproteins (i.e., αIIbβ3, α2β1, αvβ3, α5β1, and α6β1). At present, α2β1/GPVI is identified to mediate collagen or fibrin adhesion/activation of platelets, whereas αIIbβ3 mediates the adhesion of fibrinogen and the subsequent platelet–platelet interaction, a common final step of platelet aggregation shared by stimulating agonists [3,4].

The processes of platelet activation consist of the stimulation of several signal transductions and result in a shape change, thromboxane A2 formation, release reaction of adenosine diphosphate (ADP), the augmentation of coagulation activity, and finally, the exposure of latent αIIbβ3. Consequently, the plasma fibrinogen binds to the activated integrin αIIbβ3 of the platelets and leads to platelet aggregation, not only maintaining hemostasis and preventing excessive bleeding under normal conditions, but also forming a thrombus under pathological conditions, such as an endothelial injury due to an atherosclerotic lesion [5,6]. 

### 1.2. Discovery of Disintegrins from Viper Venom as Antithrombotic Agents

Disintegrin was first discovered in 1987, since trigramin, a non-enzymatic small molecular polypeptide, was isolated from the snake venom of *Trimeresurus gramineus* [7]. Trigramin inhibited platelet aggregation by blocking fibrinogen binding to aggregation agonist-stimulated platelets. At that time, it was proposed that disintegrins derived from *Agkistrodon rhodostoma*, *Agkistrodon halys*, and *T. gramineus* [8,9,10] inhibited platelet aggregation induced by various agonists, including collagen, ADP, sodium arachidonate, and epinephrine that neither affected the shape change nor the cyclic adenosine monophosphate (cAMP) level. Further studies reported that the disintegrin trigramin inhibited fibrinogen binding to ADP-stimulated platelets, and the binding ability of ^125^I-trigramin toward ADP-stimulated platelets was almost completely abolished in patients with a genetic αIIbβ3-defect disease (i.e., Glanzmann’s thrombasthenia) when compared to normal platelets, demonstrating that the fibrinogen receptor αIIbβ3 is the target of trigramin [7,11]. In addition, mAb 7E3 raised against αIIbβ3 and RGDS showed an inhibitory effect on ^125^I-tragramin binding to platelets, indicating that its binding target is αIIbβ3 and the Arg-Gly-Asp (RGD) tripeptide sequence is important for its binding activity. Importantly, the trigramin sequence clearly identified that it is an RGD-containing single polypeptide of 72 amino acid residues with six disulfide bonds [7]. Alkylated and reduced trigramin lost activity in inhibiting platelet aggregation and binding capacity toward platelets, demonstrating that the binding capacity of αIIbβ3 depends on RGD tripeptides, and its secondary structure and the disulfide bridges are essential in the expression of biological activities [7,12]. Upon intravenous administration, the bleeding time of severed mesentery arteries was significantly prolonged by trigramin, indicating that it decreases the ability of platelets to form thromboemboli in vivo [13]. Subsequently, emerging reports have shown that many disintegrins exhibit an inhibitory effect on platelet adhesion to extracorporeal circuit surfaces [14,15]. Since then, disintegrins have been thought to be potential candidates as antithrombotic agents, and this finding has motivated many pharmaceutical companies to develop a series of RGD-mimetic drugs based on the specific steric structure of the RGD loop of disintegrins. 

Among these disintegrins, a Lys-Gly-Asp (KGD) containing peptide barbourin exhibited specificity toward integrin αIIbβ3 than to αvβ3 [16]. Therefore, a cyclic KGD peptide, Integrilin, was successfully developed as an antithrombotic agent, and used clinically for the prevention of restenosis after percutaneous transluminal coronary angioplasty [17,18]. To date, three current Food and Drug Administration-approved platelet integrin antagonists, the anti-adhesive agent Eptifibatide (Integrilin, COR Therapeutics), Tirofiban (Aggrastat, Merck), and the chimeric 7E3 Fab (Abciximab, Repro) mAb raised against αIIbβ3, have been successfully developed in this field. Tirofiban is a 495-Da synthetic compound engineered to mimic the RGD sequence and acts as an anti-aggregation agent [19] and Abciximab, a mouse/human chimeric monoclonal c7E3Fab raised against integrin αIIbβ3 [20,21]. These three FDA-approved αIIbβ3 antagonists are administered intravenously. Orally active integrin antagonists have also been developed, but the clinical trials of these oral agents have resulted in increased mortality instead of beneficial effects [22]. Nevertheless, in high-risk patients undergoing percutaneous coronary intervention and transluminal coronary angioplasty, the current integrin antagonists have each demonstrated clear therapeutic benefits, as indicated by a significant decline in death rates and the reoccurrence of myocardial infarction [23].

### 1.3. Bleeding Side Effects of Current Antithrombotic Agents for Acute Coronary Syndromes

Despite the successful clinical use of integrin antagonists as therapeutic antithrombotic drugs, the risk of life-threatening bleeding directly limited their utilization in patients undergoing percutaneous coronary intervention [24]. The intrinsic mechanism of abnormal bleeding is due to the binding of αIIbβ3 antagonists to the integrin αIIbβ3, inducing conformational changes. After integrin antagonists disassociated from αIIbβ3, the conformationally changed receptor αIIbβ3 exposed the epitopes LIBSs (ligand-induced binding sites) [25]. Recent studies reported that LIBSs represent the binding sites of the intrinsic antibodies in patients administered with these drugs [26]. The complex of LIBS/intrinsic antibody recruited FcγRIIa and induced its downstream ITAM–Syk–phospholipase (PLC)γ2 signaling pathway, leading to FcγRIIa-mediated immune clearance of platelets and thrombocytopenia [27].

After bleeding, drug-induced immune thrombocytopenia and severe reactions to re-administration are the most severe side effects of αIIbβ3 antagonists [28]. These include commonly used drugs such as RGD-mimetic agents, antibiotics, and anticonvulsants, which are used to prevent in-stent thrombosis in patients undergoing percutaneous coronary intervention. Thrombocytopenia and gastrointestinal bleeding induced by RGD-mimetic drugs occur in patients administrated with αIIbβ3 antagonists [29,30]. Although the thrombocytopenia is usually resolved within two weeks after drug withdrawal, at present, there is no treatment to halt severe bleeding or rapidly upsurge the patient’s platelet count if it occurs during the thrombocytopenic phase. Given the widespread use of these efficacious drugs and the relatively high incidence, it is crucial to develop a new class of αIIbβ3 antagonists with a minimal risk of bleeding. 

### 1.4. New Insights into Antiplatelet Strategies

As established by expanded clinical trials and usage, the clinical scenarios in which the current αIIbβ3 antagonists present efficacy are more limited than originally expected. However, the important role of integrin αIIbβ3 in platelet aggregation and thrombosis remains irrefutable. Since the premise that targeting αIIbβ3 remains a fundamentally solid strategy, some investigators have looked to identify a new class of αIIbβ3 antagonists, ones that will either not cause conformational changes on dissociation or association from αIIbβ3, or reduce the occurrences of severe bleeding and thrombocytopenia in some patients. Two potential strategies have been suggested to achieve this end: targeting factors that mediate the propagation of the thrombus without affecting the initial formation of the thrombus core [31], or discovering inhibitors that bind to the extracellular domain of the integrin without causing receptor activation and conformational changes [32]. These may hold the key to safer antithrombotic therapy (Figure 1). 

#### 1.4.1. Integrin Antagonists that Minimally Affect Conformational Changes

Tirofiban, Eptifibatide, and Abciximab all cause thrombocytopenia and bleeding, which is associated with integrin conformational changes after drug dissociation. Immune thrombocytopenia occurs on the first exposure to RGD-mimetic drugs and subsequently the platelet counts decrease abruptly within hours of the commencement of drug administration [29], indicating the presence of an intrinsic antiplatelet antibody that occurs naturally. In the majority of patients with RGD-mimetic drug-induced thrombocytopenia, the binding of the antibody was drug specific and only occurred as the integrin combined with the drugs that caused the thrombocytopenia. RGD-mimetic drugs binding to the RGD recognition site of αIIbβ3 cause conformational changes and the exposure of cryptic epitopes, termed MIBSs (integrin mimetic-induced binding sites) [33], or LIBSs (ligand-induced binding sites) [34]. Recently, novel small-molecule integrin antagonists and pure disintegrins from snake venom have been developed that exhibit improved specificity and potency without causing the emergence of the LIBS epitopes of integrin β3. The small-molecule RUC-1 and the second congener RUC-2 block fibrinogen binding to activated αIIbβ3, but do not induce major conformational changes in the protein β3 subunit nor the emergence of LIBSs and presumably MIBSs [32,35], and may not be antigenic and cause immune thrombocytopenia. RUC-1 interacts with αIIb, while RUC-2 interacts with the Mg^2+^ coordinating sites of β3, termed MIDAS (metal ion-dependent adhesion site) [36]. The RGD-containing disintegrin TMV-7 from snake venom and its more potent derivatives inhibit the ligand-binding function of integrins, platelet aggregation, and thrombus formation, and importantly do not cause major conformational changes in the integrin β3 subunit or induce integrin activation [27]. 

#### 1.4.2. Integrin Antagonists that Target Integrin Outside-in Signaling

Emerging evidence has shown that mutational disruption or deletion of the talin-binding site prevented mice from thrombosis, but defected physiological hemostasis has been revealed to prolong tail bleeding times in vivo [37,38]. Since then, the inhibition of inside-out signaling has led to impairing the talin-driven integrin activation and its ligand-binding function, therefore, it is expected that these inside-out signaling inhibitors should show characteristics similar to current αIIbβ3 antagonists, which inhibit both thrombosis and hemostasis. Current conceptual advances in integrin outside-in signaling have shown the potential in developing selective inhibitors of integrin outside-in signaling as new antithrombotic drugs [31]. A permeable myristoylated ExE motif peptide was designed to selectively inhibit Gα13-mediated outside-in signaling, platelet spreading, and thrombus formation without causing excessive hemorrhage in vivo. Thus, selective inhibitors of outside-in signaling as a potential antithrombotic strategy may find the right balance between potent antithrombotic efficacy and bleeding adverse reactions to allow antithrombotic therapy with an appropriate control of bleeding risk.

### 1.5. Disintegrins as Anti-Inflammatory Agents

The immune system recognizes pathogen-related molecular patterns via a repertoire of pattern recognition receptors, among which the family of Toll-like receptors (TLRs) features prominently [39]. TLR4 recognizes lipopolysaccharide (LPS)-containing Gram-negative bacteria, while TLR2 recognizes the peptidoglycan of Gram-positive bacteria. The β2 and β3 integrins form heterodimers with an α subunit to regulate leukocyte trafficking and function [40]. Vitronectin and integrin αvβ3 contribute to the initiation of TLR2 responses to bacterial lipopeptide [41]. Our recent studies have indicated that the disintegrin rhodostomin (Rn) interacts with the αvβ3 integrin of monocytes/macrophages, leading to interference with the LPS-induced NFκB and MAPK pathways, and MyD88-dependent TLRs in the production of cytokines in phagocytes. Thus, Rn significantly represses pro-inflammatory cytokine and chemokine release, inhibits cell adhesion and migration in vitro, and elevates the survival rate of septic mice in the LPS-administration and cecal ligation puncture (CLP) model by attenuating the acute inflammatory activity caused by bacterial infections [42,43]. The capacity of Rn to reduce thrombus formation may be responsible for its antiplatelet activity through the αIIbβ3 blockade. The potential application of disintegrins, and their safer derivatives, as an inhibitor of TLR2 and TLR4 activation raises the possibility of drug development in inflammatory diseases caused by complicated microbial patterns. 

### 1.6. Disintegrins as an Anti-Tumor and Anti-Angiogenesis Agents

Angiogenesis plays a vital role in normal physiological processes such as tissue repair, embryonic development, and luteal formation [44]. Integrin αvβ3 expressed on smooth muscle cells, endothelial cells, transformed cells, and fibroblasts modulates cell migration, proliferation, and has a great impact on restenosis, tumor cell migration, angiogenesis, and atherosclerosis [45]. Several classes of integrins recognize the RGD sequence present in extracellular matrix (ECM) proteins [46], resulting in linking cytoskeletal proteins with the ECM, and are involved in bidirectional signaling that changes cellular functions. Our previous studies have reported that some disintegrins inhibit adhesion between ECMs and tumor cells via blocking αvβ3 and α5β1 integrins [47,48,49,50]. These disintegrins, as well as the synthetic RGD-containing peptides, have been shown to inhibit the experimental metastasis of melanoma cells [51,52,53]. We also found that RGD containing the antiplatelet disintegrin trifavin inhibited B16F10 melanoma cell-induced lung colonization in an experimental model [54], and trigramin inhibited the cell-substratum adhesion of human melanoma cells, and spreading on fibronectin and fibrinogen [55], as well as that the rhodostomin inhibited the ancrod-generated fibrin-triggered prostaglandin I_2_ formation of human umbilical vein endothelial cells (HUVECs) through blocking αvβ3 integrin [56,57,58]. Rhodostomin was also found to cause the cleavage of β-catenin and poly (ADP-ribose) polymerase during the apoptosis of endothelial cells [59]. However, efforts to apply αvβ3-specific RGD-mimetics in tumor therapy are still under active investigation. Further development of the recombinant venom-derived disintegrin, along with new technologies looking at additional disintegrin-like proteins with pure antagonist specificity, may offer novel therapeutic approaches in targeting tumor-induced angiogenesis

### 1.7. Novel αIIbβ3 Antagonists Derived from Disintegrins 

Life-threatening thrombocytopenia and bleeding risks limit clinical antithrombotic use in patients undergoing percutaneous coronary intervention (PCI), however, given the widespread use of these efficacious drugs and the relatively high incidence, ideal antithrombotic agents with minimal bleeding side effects are currently under active investigation. Recently, we found a unique RGD-containing disintegrin, TMV-7, purified from *Trimeresurus mucrosquamatus* venom, which, like RUC-2, binds to the domain lying between the αIIb and β_3_ subunits, mostly stabilized through strong interactions with the αIIb subunit, thus neither primes the platelets to bind the ligands nor causes a conformational change of β_3_ as identified with the ligand-induced binding site (LIBS) mAb AP5 [27]. TMV-7 has also been shown to be an efficacious antithrombotic in illumination-induced mesenteric venous thrombosis and FeCl3-induced carotid artery thrombosis models. At an efficacious antithrombotic dosage, TMV-7 did not increase bleeding risk in vivo [27]. Its unique mechanism of action may be related to targeting outside-in signaling without affecting the talin-driven inside-out signaling and platelet function (i.e., platelet fibrin adhesion and clot retraction) responsible for primary hemostasis. A recent report indicated that Gα13 and talin play vital roles in thrombin-induced αIIbβ3 bidirectional signaling and bind mutually exclusive sites of β3 cytoplasmic domain in opposing waves, demonstrating that selectively targeting integrin outside-in signaling allows for the inhibitory activity of thrombus formation, while maintaining physiological hemostasis in animal models [31,60,61]. Therefore, the elucidation of the docking structure of the disintegrin TMV-7-αIIbβ3 complex and the structure–activity relationship between TMV-7 and αIIbβ_3_ on a molecular level may provide clues for the drug development of an ideal antithrombotic R(K)GD-mimetic with a better safety profile. Furthermore, TMV-7 inhibits LPS-TLR4 ligation-mediated release of pro-inflammatory mediators by downregulating reactive oxygen species (ROS) production and FAK/NFκB/MAPK axis signaling through the blockade of integrin αVβ3 in RAW 264.7 macrophages and human THP-1 cells [62]. Furthermore, TMV-7 significantly reduced myocardial ischemia-reperfusion (I-R)-induced arrhythmias, infarct volume, as well as mortality, via downregulating apoptotic proteins Bax and Caspase-3 in rats with I-R injury, demonstrating that TMV-7 is also possibly a potent antiarrhythmic agent with cardio-protective properties [63].

### 1.8. Other Possible Applications of Disintegrins

It has been suggested that the intracellular changes during ischemia trigger the accumulation of reactive oxygen species (ROS) and metabolic intermediates. The abnormal increase of ROS is associated with oxidative stress, and implicated in ischemic stroke and acute myocardial infarction [64,65,66], resulting in potential damage to proteins, lipids, and nucleic acids. Both of these diseases are also caused by arterial thrombosis. Current clinical trials have examined a combination of fibrinolytic therapy by using αIIbβ3 antagonists and a recombinant tissue-type plasminogen activator [67,68,69], and indicate that αIIbβ3 antagonists may have a beneficial effect by reducing adverse outcomes caused by strokes, although there is a risk of increased bleeding, especially with Abciximab [69]. Therefore, whether a pure αIIbβ3 antagonist, instead of the partial antagonist, can be revaluated in this add-on therapeutic strategy on stroke treatment remains to be examined.

## 2. Conclusions

The journey of snake venom disintegrin research has been initiated since the first naturally occurring disintegrin was discovered [7] and has inspired many studies in the molecular interaction of R(K)GD-containing disintegrins with integrin αIIbβ3, αvβ3, and other integrins, leading to the fruitful discovery of potential therapeutic agents in field of inflammation, angiogenesis, tumor metastasis, and atherothrombotic diseases (i.e., acute coronary syndrome, myocardial infarction, or ischemia stroke). With the aid of advanced molecular biology techniques, and the elucidation of the physiological and pathological roles of integrins, we can utilize these disintegrins and their derivatives to target specific integrins for the further evaluation of their potential efficacy and the associated adverse reaction mechanisms. Several potential antiplatelet strategies have shown considerable promise toward preventing thrombus formation while sparing hemorrhage. These include inhibitors of the αIIbβ3 outside-in signaling [31], conformation-specific targeting of αIIbβ3 [27,32,70], and a competitive inhibitor of the fibrin D-dimer region of monomeric Glycoprotein VI [71,72]. In keeping with the concept that the differential targeting of the thrombus formation may preserve physiological hemostasis, all of these inhibitors do not seem to affect the initial adhesion of platelets in vivo and thus allow for the formation of a platelet hemostatic core in response to vascular injury. The most challenging task is how to design a safer therapeutic agent, namely a pure integrin antagonist, which specifically blocks the αIIbβ3/αvβ3 target without intrinsically activating or changing the conformation into an activated state, leading to bleeding, thrombocytopenia/activation of inflammation/angiogenesis.

The discovery of novel therapeutic antithrombotics, especially the small-mass RGD-mimetic drugs derived from naturally-occurring disintegrins, may be accelerated through clarifying the detailed ligand-receptor docking, structural biology of integrins, and ligand-integrin complex aided by X-ray crystallography. Despite the challenges of translating promising animal data into the clinic, the emergence of novel antithrombotic targets holds the promise of improving cardiovascular outcomes, minimizing iatrogenic bleeding complications, and significantly increasing the number of patients who are afforded protection by antithrombotic drugs. Likewise, these safety-improved integrin antagonists derived from disintegrins can be widely used in patients afflicted with integrin-related diseases, such as ischemia stroke, septic inflammation, angiogenesis, and tumors.

## Figures and Tables

**Figure 1 toxins-11-00372-f001:**
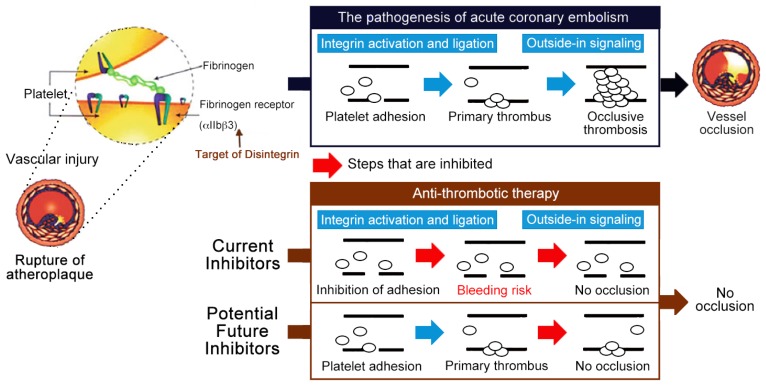
Current αIIbβ_3_ antagonists are effective antithrombotics, but have a significant bleeding risk. Schematic representation of the antithrombotic strategy selectively inhibiting outside-in signaling without causing integrin activation nor affecting the processes of primary hemostasis (i.e., platelet adhesion and fibrin clot retraction), thus these potential antithrombotics do not increase bleeding risk and have greater safety profiles than the current αIIbβ_3_ antagonists (i.e., Eptifibatide) [31].

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
