# Peer review of "From Discovery of Snake Venom Disintegrins to A Safer Therapeutic Antithrombotic Agent"

_toxins, 2019, doi:10.3390/toxins11070372_

Round 1
Reviewer 1 Report
This is generally a good and comprehensive review of this field. There still needs to be a number of English corrections to make a more elegant text. It would also make for easier reading if the authors dealt with the problem of activation by steric conformation changes to integrins by binding of inhibitors when this first comes up. They do deal with this problem several times later in the review but it would make more sense to mention this problem directly.
Author Response
We sincerely appreciate your giving us many constructive comments on our manuscript. We carefully revised our manuscript according to the raised issues. Our Ms has been also edited for English language, grammar, common technical terms, punctuation, and spelling by MDPI. We directly described the problem regarding activation by steric conformation changes to integrins by binding of inhibitors, and added the related texts in the revised MS (line 136-195).
Reviewer 2 Report
This is a highly detailed and fairly comprehensive review of the relevant literature and as such makes a useful contribution and should be published. My only (minor) concern is that the quality of the English might be slightly improved, which would increase the paper's readability - an important consideration for review papers. The issues are relatively minor, but do contribute to making the paper slightly more challenging to absorb than would be optimal. It may be worth the authors employing a professional editorial service to improve readability.
Other than this, I have no major suggestions, however the "key contribution" indicates that "Elucidation of the pathogenesis of drug-induced thrombocytopenia and bleeding may lead to development of safer antithrombotic drugs" - elucidation may be an over used word and there is a sense in which a review of this kind does "elucidate" previous findings, but another sense in which those previous studies themselves are when the "pathogenesis of drug-induced thrombocytopenia and bleeding" are in fact "elucidated" - I would consider changing the text under key contributions to avoid the implication that this paper contains original "elucidating" data (which of course should be clear from the fact that it is a review).
I began to list minor grammatical infelicities, but i quickly became apparent that there were far too many of these to individually address. In general, I would suggest the authors pay more attention to their choice of articles ("a" versus "the" etc) and, as above, suggest they contract a professional editorial service to improve readability to a level that would do justice to the impressive amount of work they have done in compiling the information present in the review.
7 Among the platelet aggregation inhibitors, disintegrins have been recognized as the unique and
Delete "the"
15 disintegrins as the tool in field of arterial thrombosis, angiogenesis, inflammation, and tumor
Replace "the" with "a"
46 Disintegrin was first discovered in 1987 since Trigramin, a non-enzymatic small molecular
Replace "since" with "when"
Author Response
We sincerely appreciate your giving us many constructive comments on our manuscript. We carefully revised our manuscript according to the raised issues. According your suggestion, we have deleted the word “elucidation”, which was used in the part "key contribution" of the previous version (line 21-24). Our Ms has been also edited for English language, grammar, common technical terms, punctuation, and spelling by MDPI.